# Reappraisal of the Concept of Accelerated Aging in Neurodegeneration and Beyond

**DOI:** 10.3390/cells12202451

**Published:** 2023-10-14

**Authors:** Yauhen Statsenko, Nik V. Kuznetsov, Daria Morozova, Katsiaryna Liaonchyk, Gillian Lylian Simiyu, Darya Smetanina, Aidar Kashapov, Sarah Meribout, Klaus Neidl-Van Gorkom, Rifat Hamoudi, Fatima Ismail, Suraiya Anjum Ansari, Bright Starling Emerald, Milos Ljubisavljevic

**Affiliations:** 1Department of Radiology, College of Medicine and Health Sciences, United Arab Emirates University, Al Ain P.O. Box 15551, United Arab Emirates; e.a.statsenko@uaeu.ac.ae (Y.S.); gl.simiyu@uaeu.ac.ae (G.L.S.); daryasm@uaeu.ac.ae (D.S.); a.kashapov@uaeu.ac.ae (A.K.); sarahmeribout@uaeu.ac.ae (S.M.); klausg@uaeu.ac.ae (K.N.-V.G.); 2ASPIRE Precision Medicine Research Institute Abu Dhabi, United Arab Emirates University, Al Ain 27272, United Arab Emirates; darsmorozova@uaeu.ac.ae (D.M.); katherine.leonchik@gmail.com (K.L.); rhamoudi@sharjah.ac.ae (R.H.); sansari@uaeu.ac.ae (S.A.A.); bsemerald@uaeu.ac.ae (B.S.E.); milos@uaeu.ac.ae (M.L.); 3Big Data Analytic Center, United Arab Emirates University, Al Ain P.O. Box 15551, United Arab Emirates; 4Department of Clinical Sciences, College of Medicine, University of Sharjah, Sharjah 27272, United Arab Emirates; 5Division of Surgery and Interventional Science, University College London, London NW3 2PS, UK; 6Department of Pediatrics, College of Medicine and Health Sciences, United Arab Emirates University, Al Ain P.O. Box 15551, United Arab Emirates; fatima.ismail@uaeu.ac.ae; 7Department of Biochemistry and Molecular Biology, College of Medicine and Health Sciences, United Arab Emirates University, Al Ain P.O. Box 15551, United Arab Emirates; 8Department of Anatomy, College of Medicine and Health Sciences, United Arab Emirates University, Al Ain P.O. Box 15551, United Arab Emirates; 9Department of Physiology, College of Medicine and Health Sciences, United Arab Emirates University, Al Ain P.O. Box 15551, United Arab Emirates

**Keywords:** aging, accelerated aging, brain aging, neurodegeneration, epigenetics, biological clocks, molecular biomarkers, rejuvenation

## Abstract

Background: Genetic and epigenetic changes, oxidative stress and inflammation influence the rate of aging, which diseases, lifestyle and environmental factors can further accelerate. In accelerated aging (AA), the biological age exceeds the chronological age. Objective: The objective of this study is to reappraise the AA concept critically, considering its weaknesses and limitations. Methods: We reviewed more than 300 recent articles dealing with the physiology of brain aging and neurodegeneration pathophysiology. Results: (1) Application of the AA concept to individual organs outside the brain is challenging as organs of different systems age at different rates. (2) There is a need to consider the deceleration of aging due to the potential use of the individual structure–functional reserves. The latter can be restored by pharmacological and/or cognitive therapy, environment, etc. (3) The AA concept lacks both standardised terminology and methodology. (4) Changes in specific molecular biomarkers (MBM) reflect aging-related processes; however, numerous MBM candidates should be validated to consolidate the AA theory. (5) The exact nature of many potential causal factors, biological outcomes and interactions between the former and the latter remain largely unclear. Conclusions: Although AA is commonly recognised as a perspective theory, it still suffers from a number of gaps and limitations that assume the necessity for an updated AA concept.

## 1. Introduction

Aging is associated with structural and physiological changes that increase the risk of developing diseases and death [1,2,3]. Aging processes are influenced by different factors, including genetic mutations, epigenetic modifications, oxidative stress and inflammation [4], resulting in the accumulation of damage and dysfunction at all biological levels [5]. Biological age (BA), also called physiological age, can be defined as the current state of an individual as a biological system, characterized by a combination of detectable life time-dependent biological parameters (determination criteria), for example, by the current profile of genomic DNA methylation, the present status of aging-associated structures in the brain, etc.

Normal aging corresponds to all monitored aging-associated processes where BA equals the chronological age. Accelerated aging (AA) is observed when the BA exceeds the chronological age and, vice versa, decelerated aging is observed when the chronological age exceeds the BA [6,7,8]. AA shares common features with normal aging, but it is also characterised by specific processes such as protein aggregation and excitotoxicity [9,10,11]. Understanding the mechanisms of aging can open opportunities for targeted therapies to slow it [9].

AA is an area of active research with unresolved issues, including the unstandardised terminology [12] and understudied mechanisms [13] including neurodegeneration (ND) described either as a type of AA [14,15] or as its outcome [14,16,17,18]. The latter view argues that certain biomarkers (BM) are ND-specific and do not detect AA [17]. Different theories have been proposed to explain the pathogenesis of AA, including genetic theory [19], the multi-proteinopathies theory [20] and mitochondrial theory [21]. Genetic theory assumes the accumulation of DNA mutations and/or gene dysregulation in AA and has certain limitations [19,22]. It considers random DNA changes but ignores chromosomal, multifactorial and monogenic alterations [13,23,24]. The multi-proteinopathies theory is based on the accumulation/aggregation of misfolded proteins leading to cell dysfunction and causing age-related diseases [20,25]. The free radical theory considers the oxidative damage to DNA and proteins by reactive oxygen species (ROS) as the primary accelerator of aging [15,26,27,28]; however, it has the problem of segregating between the normal and abnormal levels of ROS [21,29]. However, these aging theories lack reliable diagnostic BM for early identification and prognostication [20,30].

## 2. Biomolecular Aspects of Aging

Aging theories differ in their approaches to describing aging at cellular, supracellular and subcellular levels. ND aetiology can be approached with the neurocentric (NC) or neurovascular (NV) view. The latter refers to the dynamic multicellular structure called the neurovascular unit (NVU) that includes astrocytes, microglia, oligodendrocytes, precursor cells, excitatory and inhibitory neurons, endothelial cells and pericytes, and mediates the functional interactions between brain tissues per se and the blood vessels [31]. The NV hypothesis proposes that neural cells in the NVU and circulating immune cells secrete proinflammatory mediators, therefore, contributing to age-related neuroinflammation [32], cell degeneration [33,34] and endothelial impairment [34,35]. These changes disrupt molecular networks, induce damage to the blood–brain barrier [36,37] and lead to NVU dysfunction, a major cause of ND [38]. However, the exact role of NVU in ND remains unclear [39]. Accumulated evidences of the high complexity and molecular heterogeneity of the NVU network makes the search for associated BM difficult and requires whole genome studies, e.g., global transcriptome analysis followed by hierarchical data clustering [40] or single-cell/single-nucleus transcriptomics [41,42].

Molecular biomarkers (MBM) are biomolecules, their components, fragments or modifications with associated measurable parameters that serve as a tool to diagnose pathologies and monitor biological processes. MBM can be used to evaluate aging, particularly to estimate the rate of its progression [43]. Aging MBM include mRNA transcripts, proteins [44], the length of telomeres, serum markers of DNA damage [45], DNA methylation profiles [46,47], histone modifications [48,49,50,51,52,53,54,55,56,57,58,59], differentially expressed genes [42,60], non-coding RNAs [61,62,63,64] and other biomolecules (Figure 1).

Despite the large number of suggested MBM of ND, only a few of them have been validated. In most studies, the sample sizes have been too small to justify the accuracy and reproducibility of MBM data. For example, a recent study aimed at determining whether age affects different cell types in NVU resulted in the model discriminating Alzheimer’s disease (AD) from healthy control (HC) samples, revealing 15 genes related to accelerated aging (AAG): *IGF1R, MXI1, RB1, PPARA, NFE2L2, STAT5B, FOS, PRKCD, YWHAZ, HTT, MAPK9, HSPA9, SDHC, PRKDC* and *PDPK1* [42]. Of these genes, differential expression of *IGF1R, MXI1, PPARA, YWHAZ* and *MAPK9* correlate with the progression of ND and may function as facilitators or inhibitors of AD. However, questions remain on the cell-specific roles of the discovered AAGs, and their contribution to AD pathogenesis and interactions in NVU. Moreover, the study cohort included only 11 AD patients and seven HC, which is insufficient for justifying AAGs as MBM in AD [42]. ND results from multiple structural changes at different genetic loci over a period of time [65,66]. AD represents 90% of ND cases, and the chances of it developing increase with changes in the 15 indicated genes predisposing to ND (NDG): *GBA1, APP, PSEN1, MAPT, GRN, SETX, SPAST, CSF1R, C9orf72* [67], *TET2* [68], *TBK1* [69], *TOMM40, APOC1* [70], *APOE* [70,71] and *TREM2* [72,73,74,75,76,77,78]. Surprisingly, there is no overlap between both sets of genes resulting from different studies: AAGs and NDGs. In dementia with Lewy bodies, multi-cognitive decline and corticobasal degeneration, the risk factors shown to be involved are the *APOE* e4 allele and the mutation spectrum for *TREM2* gene [78,79,80,81,82,83,84,85,86,87].

DNA methylation rate reflects the rate of aging. Approximately 1.5% of genomic DNA contains 5-methylcytosine (5-mC) that decreases during ontogenesis [88]. The level of 5-mC is found to be highest in embryos and it decreases gradually with age [89,90]. Global genomic DNA hypomethylation in aging proceeds along with hypermethylation of CpG islands (CGIs) in the mammalian genome where 60% of CGIs are associated with gene promoters and involved in the regulation of gene transcription [91]. Changes in DNA methylation patterns known as “epigenetic drift” are associated with aging across the entire lifespan [92].

Age-predictive models demonstrate gradual linear changes in the DNA methylation profile in normal aging; however, environmental or genetic risk factors accelerate aging [93]. In monozygotic twins, the divergence of the methylome increases at different rates [94]. Furthermore, the DNA methylation profile was proposed as mechanisms of the epigenetic clock [95,96,97] by analogy with the biological clock [98,99]. Monitoring deviations between biological and chronological age helps to study development and aging across the lifespan [100]. Horvath [101], Hannum [93] and PhenoAge [102] epigenetic clocks serve as markers of ND [102,103,104,105,106], with the first of these showing the strongest correlation between epigenetics and chronological age [107]. 

Histone modifications can be used as potential aging MBM; however, the heterogeneity of the studies and the animal models limited the applicability of the findings. For highly abundant transcription activation mark H3K4me3 [48], the decrease in its level was shown to correlate with an extended lifespan in *Caenorhabditis elegans* [49], while an increase has been linked with AA and a reduced lifespan in *Drosophila melanogaster* [50]. For the heterochromatin-associated histone transcription repression mark H3K9me3 that is gradually lost during human and mouse aging in haematopoietic stem cells [51], the most significant changes occurred in the repressive regions in *C. elegans* [52] and models of AA [53]. The role of the repressive histone mark H3K27me3 associated with transcriptional silencing [54] in aging is controversial, as studies have shown both an increase and a decrease in its level during aging [55,56,57,58,59].

Increased H4K20me3 and H3K4me3 and decreased H3K9me1 and H3K27me3 histone modification levels have been described as common age-associated epigenetic marks [108,109,110]. In AD, an increase in the gene activation-related histone mark H3K4me3 was shown in both the CK-p25 tauopathy mouse model and in the hippocampus of AD patients [111,112]. Other histone methylation marks identified in AD patients’ brain included: H4K20me2, H3K4me2, H3K27me3, H3K79me1, H3K79me2, H3K36me2, H4K20me3, H3K27me1 and H3K56me1 [113,114]. In addition to methylation, histone acetylation marks H3K9ac, H3K14ac and H4K16ac have been shown to be associated with aging and AD [110,111,113,114,115]. Among other ND-associated histone modifications in AD, histone phosphorylation H4S47p and H3S10p {chaput2016potential} and histone ubiquitination H2BK120ub were reported [114,116,117]. The exact regulatory modes of both the modifications and mechanisms of their interaction and interplay with other factors in such complex processes as aging and ND are still unclear and require further systematic research.

Non-coding RNAs (ncRNAs) are used as aging MBM [118,119,120,121]. Long non-coding (lncRNA), for example, the growth-arrest-specific transcript 5 (*GAS5*), plays a significant role in cell proliferation and apoptosis [122,123,124], and its down-regulation leads to the phosphorylation of the tau protein in ND [125,126]. Long intergenic brain cytoplasmic RNA 1 (*BCYRN1*) expressed in the dendritic domains of neurons is down-regulated in aging [127].

*MicroRNAs* (miRNAs) mediate brain aging through the regulation of gene expression, impact neuronal plasticity and influence tau protein metabolism [128,129,130,131,132,133,134,135,136,137]. The regulation of *MiR145a* and *MiR-375* was shown to be age-dependent in mouse brains [138,139,140]. The MIR29 family, *MIR339-5p, MIR195* and *MIR107*, regulate the expression of beta-secretase 1 that is responsible for proteolysis of the amyloid precursor protein [141,142,143,144,145,146]. Interestingly, *Mir34* was shown to play a protective role in Drosophila [147], and *MIR144/MIR451* was found to regulate ADAM metallopeptidase domain 10 in AD [148]. Over 20 miRNAs secreted into the cerebrospinal fluid by hypothalamic stem cells were found to control the aging rate in mice [149], which is especially important as the hypothalamus was placed at the base of human brain aging [150]. As most miRNA studies have been conducted on non-human models, their relevance to human data needs to be verified; hence, future studies will be required to define the roles of miRNAs [151].

Circular *RNAs* (circRNAs) are a recently described type of ncRNA with age-dependent expression in skeletal muscles [152] and an abundance in the brain [153], playing a role in ND through their interaction with miRNAs. For example, *CIRS-7* potentially functions as a sponge for *MIR7-1* [154] and its level is dramatically reduced in the AD brain [155]. Cerebral circRNAs are associated with neurotransmitter function, synaptic activities and neuronal maturation and target the expression and availability of specific age-related mRNAs in the brain. At least four circRNAs were found to be involved in postoperative neurocognitive disorders [156]. Another study revealed nearly 1200 cerebral circRNAs in a rat aging model [157]; however, these circRNAs still await their complete characterisation.

## 3. Aging of Organs and Systems beyond Neurodegeneration

Aging affects organs and systems with different rates of change; therefore, the AA concept needs to be adjusted when applied to individual organs. For example, ovarian aging implies a loss of follicle numbers and decreased oocyte viability. Typically, an accelerated decline in fertility begins around the age of 38 years and continues until the climacteric [158]; however, a non-uniform decrease in follicle numbers results in a large variation in menopause onset. The BA of the male reproduction system can also be assessed by fertility, but the arrest of reproductive capacity is reversible in older men, with lifestyle and disease factors prevailing over other determinants of aging [159]. In mice, oxidative stress, inflammation, DNA damage and de novo mutations accelerate testicular aging [160,161,162,163], while enhancing antioxidant enzyme activities with growth differentiation factor 11 protects the testes [164]. A progressive age-related drop in Leydig and Sertoli cell function [165], testicular size [166] or testosterone levels was demonstrated in older elite men [167]; however, no decrease in testicle size or the levels of testosterone was observed in a cohort of older elite men without comorbidities [168,169]. 

The cumulative effect of disease rather than age may account for changes in the male reproductive system throughout life. Obviously, chronological age does not always determine reproductive BA, and adopting the AA concept to the reproductive system is not allowed, which illustrates the challenges in assessing BA at the organ and system levels.

Sex hormones that affect fertility are part of the endocrine system. Data on the susceptibility of the endocrine system and metabolism to aging differ between organs and sex. The hypothalamic–pituitary–testicular axis in men does not undergo dramatic chronobiological changes, accounting for only 35–50% of men over 80 with reduced testosterone levels [170,171]. Conversely, diabetes mellitus and obesity predispose to accelerated adipose tissue dysfunction, affecting telomere length [172,173]. Adrenal and thyroid functions undergo less prominent age-related changes than their hypothalamic regulation [174]; therefore, assessing BA from hormonal findings is challenging. With aging, hormone activity decreases and endocrine alterations are established [175]. BA is affected by the level of glycosylated haemoglobin, glucose, triglycerides, and low-density and total cholesterol [176,177,178]. The modulation of these parameters, lifestyle and environmental factors can prevent or contribute to AA [179]. The effectiveness of hormone replacement therapy for aging reversal is questionable though [180].

Environmental and endocrinological factors affect the BA of connective tissue. The status of the skeletal system reflects the individual endocrine profile and micronutrient balance [181,182,183] as well as environmental and occupational attainments [184]. For example, bone resorption in astronauts prevails over its formation due to the effects of microgravity; however, bone density normalises after the flight [185,186]. Skin elasticity serves as a marker of aging, whose rate can be modified due to estrogen deficiency, metabolic alterations and exogenous factors (burns) [187,188,189,190,191]. Fibroblasts constitute a natural cell stock that allows skin rejuvenation, repair and decelerated aging [192]. In connective tissue, a combinatory effect of internal and external factors determines BA more accurately than the chronological one [193,194]. Therefore, the inability to account for the decreased aging rate reveals the weakness of the AA concept.

Studies in other systems have also reported the reversibility of age-related changes in them. For example, physical training can rejuvenate the respiratory system by expanding the alveolar space. However, studies on these issues did not comprehensively evaluate the BA of the lung since the impact of muscle atrophy on the results in the spirometry test was not considered [195,196,197]. Lifestyle changes (e.g., calorie restriction and physical activity) could also reverse aging in patients with early stages of chronic kidney disease [198]. Another example is shown by the discovered potential to rejuvenate the kidneys with up-regulation of the *Klotho* gene [199]. These evidences speak for a limited generalisation of the AA concept. AA affects various systems and cross-organ communication. The interaction between systems can impede the atrophy of an organ through compensatory mechanisms in other organs. Several studies have demonstrated the role of the central nervous system in reversing the aging of other systems and organs [200,201,202]. Endocrine and cardiovascular diseases promote renal aging [203]. Conversely, kidney transplantation can revive other parts of the body [204,205]. The characterisation of organ- and system-specific aging processes is challenging and will require combinatory approaches that are largely missing in the AA concept.

## 4. Limitations of the AA Concept

The AA concept is closely linked with and needs to be considered within the context of individual capacities and personalised structure–functional reserve mechanisms (Figure 2). The generalized term “structure-functional reserves” is introduced to approach the observed variability in multiple structure–functional parameters at different levels (expression of genetically and epigenetically regulated genes, number, viability and functionality of cells, number of synapses and intercellular contacts, secretion of cytokines, potency of physiological responses, etc.) in a population or group of subjects. The description of this term could be further developed and linked to the statistical distributions of measurable biological parameters in the population and to the norm of reaction resulting from the influence of environmental factors on trait variation.

Physiological reserves reflect the remaining capacity of an organ to perform its function. Aging and diseases lead to atrophy, reducing the number of cells and supracellular structures [206,207]. In the context of brain aging, its physiological cognitive reserve is assumed by the level of education, occupational and environmental attainments and the performance of cognitive tests [206]. Reversible forms of mild cognitive impairment (MCI) and dementia represent clinical examples of restoring individual reserve potential that are not in line with AA theory [208,209]. Neural compensation in the elderly leads to the formation of secondary brain networks [210], which decelerate the aging of the brain [206,211]. Reversion to MCI in elderly patients has been reported to be as a result of specific lifestyle activities and cognitive stimulation throughout life [212,213].

Appraising age requires an accurate estimation of individual reserves that account for biological and chronological age differences. In neuroscience, machine learning models establish an association between the number of years lived in good health and brain-imaging data with an accuracy of 2.1–4.9 years [214,215]. An individual’s brain age can also be calculated as the difference between chronological age and the predicted BA [216]. In obstetrics, the evaluation of gynaecological status takes into account both reproductive health and potential fertility. Therefore, an overall BA depends on the reserve capacities of individual systems and organs [217,218].

Variance in individual reserve capacities complicates the precise assessment of BA. Brain AA criteria are unclear since normal aging indicators are still missing [219], and no reference curves for brain changes have been created yet. Moreover, the rate of aging is hard to assess due to the lack of a standardised methodology that could take into account individual reserve potentials. Furthermore, methodological discrepancies lead to contradictory findings between different studies that vary between laboratories. For example, published studies report that AD results in the addition of 1.5 years to the brain age; MCI adds 1 year; multiple sclerosis (MS), 0.41 years; Parkinson’s disease (PD), 3 years; and schizophrenia, 5.5 years. However, the last two pathologies impact cognition in a milder and slower way than AD [220,221,222]. Other studies have reported an added brain age of between 6 and 9 years in AD, which is more consistent with the findings on MCI and PD [223]. 

The weaknesses and limitations of certain studies show the need for caution when assessing results. For example, the studies on age-related brain atrophy commonly have a cross-sectional design that is less accurate compared to the longitudinal one [224]. Many studies are based on small non-representative cohorts [225,226,227]; therefore, the usage of the resulting mathematical models is low. Another challenge for the AA concept is due to the focus on middle-aged adults and elderly patients in certain brain aging studies. These studies should not ignore individual prenatal pathologies and childhood trauma affecting the brain’s health and BA [228]. Applying the concept of AA to localised degeneration is complicated since different parts of the brain age unevenly [229]. For example, in localised ND, the BA assessment reflects the level of damage to the most vulnerable brain parts (e.g., substancia nigra and ruber nuclei in PD) [230,231,232]; however, one should also consider the brain resources that can minimise the atrophy effects [233]. In systematic ND, the brain ages faster than within localised ND [234,235], showing an apparent difference in the speed of atrophic changes [236].

Contemporary neuroscience lacks a clear explanation of the interaction between different causal factors due to the polyetiological nature of ND. It is still unclear whether chronic diseases lead to or result from ND [237,238] since the genetic, environmental and lifestyle factors interact in an undefined way [18,239,240]. Several articles have revealed a misalignment between dementia risk, cognitive performance and MBM levels [241,242]. Another disadvantage of AA studies is the inability to account for the influence of medications used by participants on study results [243]. Last but not least, AA represents a diagnostic but not pathognomonic signature in ND and in psychiatric diseases: schizophrenia, bipolar disorder and major depressive disorder [222,244,245], since the entire range of symptoms observed in these patients cannot be explained by brain aging only [222,246,247].

Finally, drug therapy could extend reserve abilities, for instance, in psychiatric conditions related to MCI and brain AA. Antidepressant medication has been reported to help convert MCI to true reversible MCI [248]. Certain cognitive disorders have demonstrated a reversible pattern in cognitive performance upon treatment [248,249]. Sex hormone replacement in ND reduced the risk, delayed the onset and slowed the progression of cognitive impairment [250]. Antioxidant-based therapy also alleviated the severity of the disease [251,252]. Recent ND studies have described a number of novel therapeutic options including specific antibodies, inducers of cell proliferation and NAD+ supplementation that are able to target mitophagy, protein aggregation and cellular senescence [9,18,253]. The observed treatment effects question the irreversible changes claimed by the AA concept.

## 5. Recommendations for Further Development and Improvement of AA Concept

### 5.1. Statistical Models

Building highly accurate machine learning models is the most common solution to distinguishing AA from normal aging. Recent articles suggest a set of approaches for improving the quality of these statistical solutions by reviewing the research data on MBM, the identification of potential gaps or inconsistencies, the incorporation of additional data sources or adjusting of research methodology, and the conduction of more rigorous statistical analyses, which will reveal any trends or correlations in the data that may have been overlooked. These suggestions have been commonly made by peers and experts who can provide valuable insights and suggestions. The following is the list of parameters that should be controlled to improve the diagnostic model of accelerated aging based on the use of specific molecular biomarkers:○*Sample size*. The number of individuals in the study can affect the statistical power of the analysis, and larger sample sizes generally provide more robust results.○*Biomarker types*. The choice of biomarkers can impact the diagnostic model used, as different types of biomarkers may require different statistical analyses.○*Age range*. The age range of the study population can influence the types of biomarkers identified, as some biomarkers may be more prevalent in certain age groups.○*Data normalization*. Normalization of the data is critical to ensure that data collection or processing differences do not affect the analysis.○*Statistical methods*. The choice of statistical methods used can impact the sensitivity and specificity of the analysis, and different methods may be more appropriate for different types of data.

Careful consideration of these parameters will be critical in developing diagnostic models based on specific molecular biomarkers of accelerated aging. Recommendations on future developments of diagnostic models also include the functional characterisation of organ-tissue-specific, single-cell-specific and disease-specific molecular clocks, the integration of epigenetics into diverse large longitudinal studies, the exploration of additional epigenomic marks of aging, and the establishing and generation of data in robust non-human aging models.

### 5.2. Molecular Clocks

Consideration of organ-specific and tissue-specific aging molecular clocks in non-human models would further resolve the complexity of aging processes. Several useful molecular clocks in mice have already been reported [254], including those specific to organs and tissues: the liver [255,256,257], lung [255,256], blood [256,258], heart and cortex [255], adipose, kidney, muscle [256], and multi-tissue [259].

### 5.3. Single-Cell Epigenomics

The study of aging at single-cell resolution represents an important direction in aging science. In particular, the variation in gene expression and the corresponding changes in aging tissues and organs [260,261] suggest that single-cell methods will be needed for a detailed analysis of accelerated aging. For instance, a rise in cell-to-cell variability with age (“epigenomic noise”) associated with methylation increase in both H3K4me3 and H3K27me3 and with transcriptional heterogeneity was shown in immune cells in blood [262] and in muscle stem cells [263], respectively. Although arranging epigenetic clocks at the single-cell level will be technically challenging, new emerging methods [264,265] and deep-learning-based computer algorithms [266,267,268] may help to construct them.

### 5.4. New Epigenetic Biomarkers

The search for new epigenetic marks of aging represents another challenge and opens up new exciting research opportunities. The connections between aging and DNA modifications other than methylation are puzzling. Nevertheless, the signs of such associations are obvious. For example, deregulation of histone H4 acetylation (H4K12) [269] and accumulation of histone variant H2A.Z [270] with age were observed in a mouse hippocampus. Changes related to the aging-related acetylation of H3 (H3K9ac) and H4 (H4K16ac) histones were found in the brain of AD patients [113,271]. Furthermore, investigations on the modulation of histone acetylation by SIRT6 HDAC linked to longevity in mammals could lead to potential pharmacological developments to target AA [272,273,274]. 

### 5.5. Consideration of Ageotypes

Diagnostic models can be adjusted to different personalised physiological subsets of aging, ageotypes [44]. This approach considers various factors such as genetics, molecular clock parameters, other epigenetic changes, lifestyle habits and environmental exposures that may influence individual aging rates. Researchers can further adjust the diagnostic model of accelerated aging by incorporating numerous potential biomarkers of aging and health metrics [44,275,276] to more accurately classify ageotypes and monitor the effectiveness of future interventions on each subset.

### 5.6. Genetic Predisposition to AA

Certain diseases can be considered as clinically important models of human genetic predisposition to AA. In particular, the phenotypes of genome instability disorders that result from autosomal recessive mutations are associated with ineffective genome maintenance systems, deficiencies in DNA helicase activity or an aberrant nuclear architecture. Three groups of these disorders include the sunlight hypersensitivity disorders: *Xeroderma pigmentosum* (XP), Cockayne syndrome (CS) and trichothiodystrophy; the ionizing radiation hypersensitivity disorders including Ataxia telangiectasia (AT) and Nijmegen breakage syndrome (NBS); and the progeroid disorders including Werner syndrome, Hutchinson–Gilford progeria syndrome, Bloom syndrome, Rothmund–Thompson syndrome and Fanconi anemia [277,278,279].

### 5.7. Application of AA Animal Models

The emergence of new animal models exhibiting aging-related features (accelerated senescence, damage of nuclear envelope, increased accumulation of genomic lesions) can significantly contribute to AA research [280]. Well-developed mouse aging models provide the possibility of testing interventions and modulators [254]. For example, acceleration of the epigenetic clock by a high-fat diet, and the effects of caloric restriction and rapamycin were demonstrated in mice models [255,257]. Emerging new animal aging models include a vertebrate with the shortest captive lifespan, killifish (*Nothobranchius furzeri*) [281,282,283,284,285,286,287], longevity models such as naked mole rats (*Heterocephalus glaber, Fukomys mechowii*) [288,289,290], Brandt’s bat (*Myotis brandtii*) [291,292,293], olm (*Proteus anguinus*) [294,295,296], bivalve *(Arctica islandica)* [297,298] and non-aging organisms: Hydra *(Hydra vulgaris*/*Hydra magnipapillata*) [299,300,301,302] and Planaria *(Schmidtea mediterranea*) [303,304,305]. The use of these models can provide the generation of new robust aging data.

## 6. Conclusions

i.The concept of an increased rate of age-related changes has certain weaknesses and limitations that are considered in the current review. In particular, so far, no unified methodology and terminology has been established in the field. The studies that justify the AA concept have too low sample sizes. Some age-related changes appear to be reversible under certain conditions.ii.Aging MBM help to estimate the aging rate increase due to a developed pathology or the exhaustion of individual reserves. A large variety of MBM candidates in different combinations can be associated with the aging brain; however, their validation, clinical interpretation and use in disease subtyping remain a challenge.iii.Activation of the regenerative mechanisms, and restoring metabolic and energy molecular reserves with novel therapeutic options could be potential ways to decelerate aging in the CNS. For example, sex hormone replacement, antioxidant-based and target therapy, and environmental and lifestyle factors’ improvement may delay ND. Future longitudinal studies could provide clinics and society with more options to prevent AA and slow the aging rate.

## 7. Afterword: Aging Science History and Theories

Several theories have been postulated to explain the possible biological meaning or evolutionary role of aging [306]: the evolutionary advantage of the species (1890s, Weisman); the accumulated mutation theory (1952, Medawar); the antagonistic pleiotropy (1957, Williams); the replicative senescence (1965, Hayflick); and the disposable soma theory (1972, Kirkwood). Furthermore, the causative theories of aging can be arranged in two groups: (I) Genetic (programmed) and (II) Stochastic (damage) theories. Programmed theories include programmed longevity theory, endocrine theory and immunological theory. Stochastic theories include wear-and-tear theory, rate of living theory, cross-linking theory, free radicals theory and somatic DNA damage theory [307].

Finally, the theories of aging can be classified by biological level and divided into: molecular level theories including gene regulation, codon restriction, error catastrophe, somatic mutation and dysdifferentiation theories; cellular level theories including cellular senescence–telomere theory, free radical theory, wear-and-tear theory and apoptosis theory; and system level theories including neuroendocrine theory, immunologic theory and rate of living theory [308].

The first publications of AA experiments performed at the Laboratories of the Rockefeller Institute for Medical Research date back to the 1920s with “normal aging” and “rate of aging” terms applied to the effects of light on Drosophila inbred in the dark [309,310]. Since then, the numbers of references on “aging”, “aging rate” and “AA” has reached 614,132; 56,088 and 21,401, respectively [311].

In 1928, the Professor of Neurology of the Columbia University, Frederick Tilney, published the work “The aging of the human brain”, where, in particular, the AD patient brain was compared to the normally aged one in the diagnostic context of the number of plaques. In this work, he also stated the abundance of senile plaques in all human brains after the age of 90 years, the influence of unfavourable factors and diseases on the brain and the importance of aging brain research. Recorded a century ago, his words are worth repeating today: “It is amazing how little general or particular interest man has shown in the most important organ of his body and life. Up to the present time he has devoted relatively little attention and much less capital to the understanding of that part of his machinery which is the secret of his success and the only hope for his future progress, if not his actual salvation… The ridiculous stupidity of annually consecrating appalling sums of money to the savage purposes of destruction should in time shock human intelligence out of patronizing such futilities and into wiser realizations. Certainly, one liberally supported and effective brain institute would prove an incomparably more profitable investment for civilization than the most powerful battle fleet that ever sailed the seas.” (Tilney, 1928 [312]).

## Figures and Tables

**Figure 1 cells-12-02451-f001:**
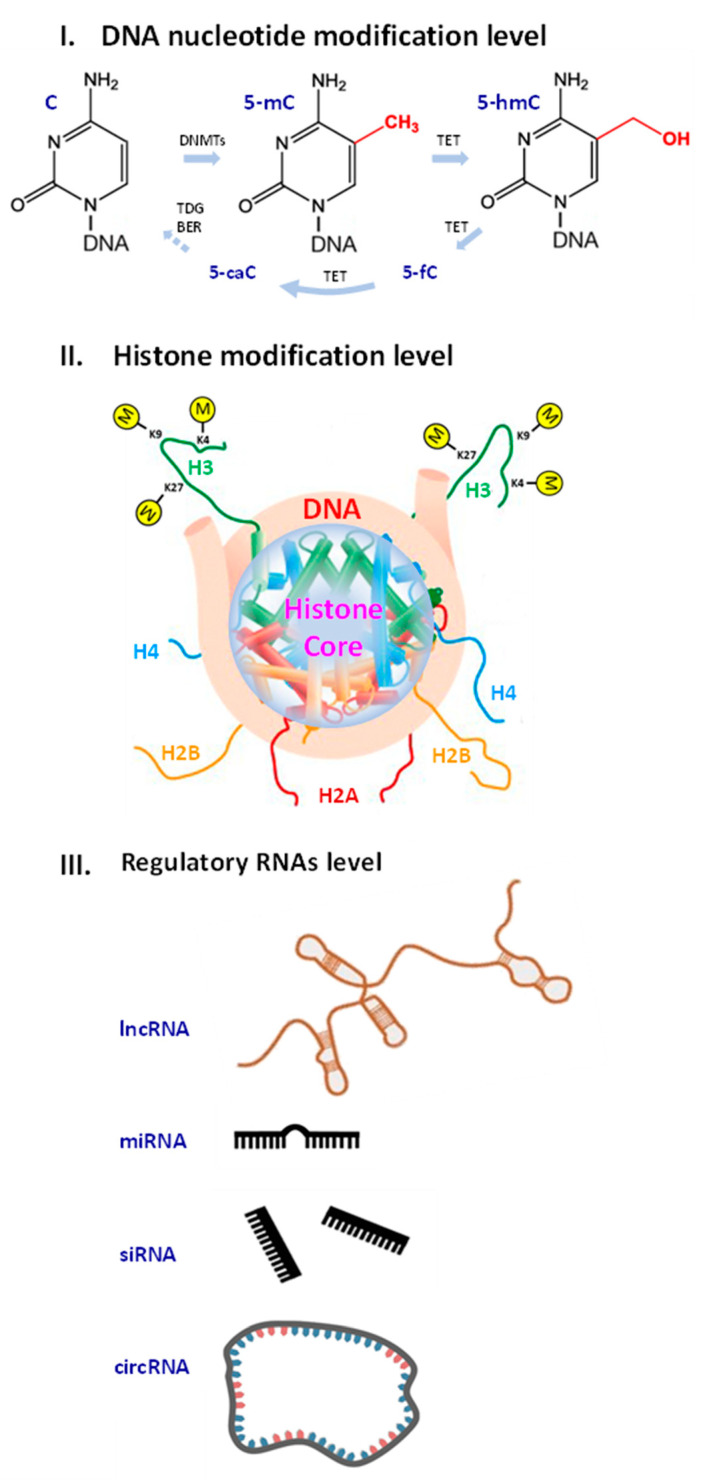
Examples of epigenetic changes and factors affected by accelerated aging at different levels that could serve as potential biomarkers of aging. (**I**) DNA nucleotide modification level: cytosine (C) can be converted to 5-methylcytosine (5-mC) and oxidised further to 5-hydroxymethylcytosine (5-hmC). (**II**) Histone modification level: Three examples of histone 3 (H3) lysine methylation marks are indicated. Lysine 4 (K4) H3K4me3 mark is associated with active genes, lysine 9 (K9) H3K9me3 is an established mark of transcriptionally repressed heterochromatin and lysine 27 (K27) H3K27me3 mark is linked to both transcriptional activation and repression. (**III**) Regulatory RNAs level: several types of ncRNAs, long non-coding RNA (lnc RNA), microRNA (miRNA), small interfering RNA (siRNA), circular RNA (circRNA), shown schematically.

**Figure 2 cells-12-02451-f002:**
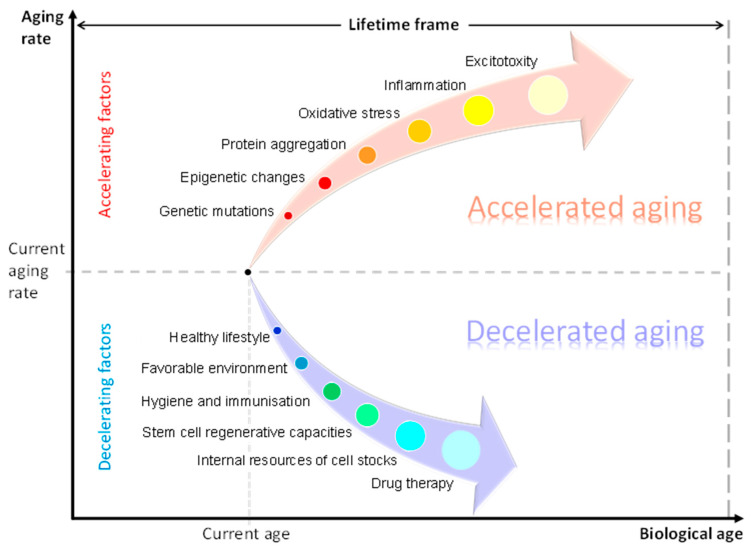
Factors influencing the rate of aging Factors accelerating aging include: genetic mutations, epigenetic changes, protein aggregation, oxidative stress, inflammation and excitotoxity. Factors decelerating aging include: a healthy lifestyle, favourable environment, hygiene and immunisation, stem cell regenerative capacities, internal resources of cell stocks and drug therapy.

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
