# Peer review of "Reappraisal of the Concept of Accelerated Aging in Neurodegeneration and Beyond"

_cells, 2023, doi:10.3390/cells12202451_

Round 1
Reviewer 1 Report
Please find below my comments and suggestions.
Line 30. “individual structure-functional reserves”. Please explain the meaning of this sentence
Line 47 “biological age”- please provide a definition of biological age. What are the criteria determining biological age?
Line 52. “including ND described either as” please define the abbreviation
Line 55. “psychogenesis of AA” what does it mean?
Line 61. “A limitation of this theory lies in the difficulty of phenotyping the diseases”. Please explain. What do you mean? There are several well-established approaches to detect and study protein misfolding and proteostasis.
Line 66. There are many other theories of ageing, please list them all.
Line 73. “The NV hypothesis proposes that neural cells in the NVU secrete proinflammatory mediators”. Only neural cells? What about immune cells and others?
Line 79. “NVU network makes the search for associated BMs difficult and requires whole genome studies”. Please explain, how the whole genome studies ()and what type of whole genome studies? whole genome sequencing? Transcriptome analysis?) may help with identification of such BMs. It was stated previously in the text that BMs abbreviation refers to genetic biomarkers. Why do the authors focus on genetic biomarkers only and ignore a plethora of other biomolecules known to be involved in ageing?
Line 129. “For example, a recent study aimed at determining whether age affects different cell types in NVU resulted in the model discriminating Alzheimer’s disease (AD) from healthy control (HC) samples”. Why do the authors focus on one study (especially, given the very small cohort size) and ignore many others on the same subject? I suggest removing the table 1 from the text.
Line 139. “GF1R, MXI1, PPARA, YWHAZ” All names of human genes should be in italic, not only in this particular sentence, but throughout the text.
Line 161. “Histone modifications as potential aging MBMs, however, the heterogeneity of the
studies and the animal models limited the applicability of the findings”. Perhaps several words are missing in this sentence, please double check. Perhaps authors tried to say “Histone modifications can be used as potential” etc
When discussing the role of histone modifications in ageing, authors focus on C. Elegans and Drofophila and their “normal ageing”, instead of discussing AA in humans in relation to neurodegeneration and neurological diseases. This sub-chapter should be re-written and significantly expanded, with focus on AA in humans in relation to neurodegeneration and neurological diseases, and impact of histone modifications on aforementioned processes.
Line 200. AGING OF INDIVIDUAL ORGANS
Here, the authors spend a great deal of time discussing ageing of female and male reproductive system, connective tissue, and respiratory system. They also discuss in detail the role of the sex hormones in the ageing of particular tissues.
Surprisingly, they say nothing about normal and accelerated ageing of the central nervous system and peripheral nervous system. Is not it the topic of their review? What is the impact of sex hormones on ageing of nervous system? On AA? And so on and so forth
Line 263. “For instance, the number of neurons and synapses reflects the brain’s structure-functional cell reserve, the number of follicles - the ovarian one, etc.” Are the authors trying to say that number of the neurons steadily decreases with age in a way the number of follicles does? In such case the statement is wrong. With ageing, the total number of neurons is rather non-significant (see, for example, https://www.ncbi.nlm.nih.gov/pmc/articles/PMC4366680/
Furthermore, what about numerous animal models of AA? (https://www.ncbi.nlm.nih.gov/pmc/articles/PMC10186612/, etc, etc)
What about genetic predisposition to AA? And so on and so forth.
In my opinion, despite impressive list of the articles quoted, the manuscript still has a room for improvement and can not be accepted in its current form. Despite being entitled "Reappraisal of the concept of accelerated aging
in neurodegeneration and beyond" the review omits the key studies (and approaches to study) on accelerated ageing and neurodegeneration.
Reviewer 2 Report
The authors apparently reviewed large number of papers in aging research and attempted to appraise the concept of accelerated aging (AA). The information after the analysis is useful to general readers but some aspects of the papers could be better presented.
1. The main topic is about the AA, but there is no systemic and detailed description of AA to behin with the paper. So the authors need to elaborate the concept of AA in more details. The main papers and the history of the theory development. Also, please detail what is the definition of normal aging as to compare to AA.
2. Another issue is the lack of clarity regarding the impact of certain biological processes on aging. Are they negative or positive? For example, the DNA methylation on aging process. For readers not familiar with the methylation and aging, it is confusing to read the part of methylation (line 151). Please at least make it clear whether methylation rate corresponding to aging positively or negatively.
3. The authors also touched telomer shortening in the section about methylation, which add another layer of confusion. Therefore, the authors should carefully read the paper many times by themselves and will see the abruptness, sudden change of topic, random insertion of a seemingly unrelated sentence in the writing in many places. Correction to this issue will make the paper read much more smoothly.
4. Why some genes are associated with references, but many others are not, with refs blank in Table1?
Round 2
Reviewer 1 Report
The article can be published in its current form as all my comments/concerns have been addressed
Reviewer 2 Report
The authors made efforts revising the original manuscript and the resultant revision improved its readability and clarity. No more issues.